# Quantification of Wnt3a, Wnt5a and Wnt16 Binding to Multiple Frizzleds Under Physiological Conditions Using NanoBit/BRET

**DOI:** 10.3390/cells14110810

**Published:** 2025-05-30

**Authors:** Janine Wesslowski, Sadia Safi, Michelle Rottmann, Melanie Rothley, Gary Davidson

**Affiliations:** Institute of Biological and Chemical Systems-Functional Molecular Systems (IBCS-FMS), Karlsruhe Institute of Technology (KIT), 76131 Karlsruhe, Germany; janine.wesslowski@kit.edu (J.W.); sadia.safi@kit.edu (S.S.); michelle.rottmann@student.kit.edu (M.R.); melanie.rothley@kit.edu (M.R.)

**Keywords:** Wnt signaling, ligand–receptor binding, Wnt-FZD binding, NanoBiT/BRET, HiBiT, LRP6, GFP-Wnt, HiBiT-FZD

## Abstract

Upon engagement of one of the nineteen secreted Wnt signaling proteins with one of the ten Frizzled transmembrane Wnt receptors (FZD_1–10_), a wide variety of cellular Wnt signaling responses can be elicited, the selectivity of which depends on the following: (1) the specific Wnt-FZD pairing, (2) the participation of Wnt co-receptors and (3) the cellular context. Co-receptors play a pivotal role in guiding the specificity of Wnt signaling, most notably between β-catenin-dependent and -independent pathways, where co-receptors such as LRP5/6 and ROR1/2/PTK7 play major roles, respectively. It remains less understood how specific Wnt/FZD combinations contribute to the selectivity of downstream Wnt signaling, and we lack accurate comparative data on their binding properties under physiological conditions. Here, using fluorescently tagged Wnt3a, Wnt5a and Wnt16 proteins and cell lines expressing HiBiT-tagged Frizzled, we build on our ongoing efforts to provide a complete overview of the biophysical properties of all Wnt/FZD interactions using full-length proteins. Our real-time NanoBRET analysis using living cells expressing low receptor levels provides more accurate quantification of binding and will help us understand how these binary engagements control Wnt signaling outputs. We also provide evidence that LRP6 regulates the binding affinity of Wnt/FZD interactions in the trimeric Wnt-FZD-LRP6 complex.

## 1. Introduction

Wnt signaling is initiated and controlled at the cell membrane through the interaction of 19 secreted Wnt lipo-glycoproteins (ligands), 10 principal Wnt receptors called Frizzled (FZD_1–10_) and several Wnt co-receptors such as LRP5/6, ROR1/2, PTK7 and RYK [1,2,3,4,5]. The intercellular Wnt signals transduced from this large repertoire of possible ligand–receptor interactions are diverse and regulate a multitude of developmental processes as well as tissue homeostasis in adults [1,2,6]. The biophysical mechanisms that select pairing of ligand–receptor interactions must therefore play a key role in specifying Wnt signaling, and much effort has been placed on studying the relative strengths of Wnt-FZD interactions. The first evidence that differential Wnt/FZD binding affinities regulate Wnt signaling emerged from *Drosophila* studies, where the stronger signaling of Dfz2 compared to Dfz1 correlated with a tenfold increase in binding affinity for Wg [7]. Additionally, an earlier study revealed that two mammalian FZD family members (FZD_3_ and FZD_6_) could not bind Wg, highlighting the divergent nature of Wnt/FZD interactions and the non-canonical properties associated with these particular two Frizzleds [8]. Indeed, some Wnts preferentially activate non-canonical (β-catenin-independent) or canonical (β-catenin-dependent) signaling. Canonical ligands include Wnt1, Wnt3a, Wnt7a and Wnt8, whereas non-canonical ligands include Wnt2, Wnt4, Wnt5a, Wnt7b and Wnt11 [1,9,10]. It is important to note, however, that there are no strict preferences for individual Wnts because promiscuous effects arise due to different cellular contexts, e.g., because of differences in the availability of Wnt receptors and co-receptors [11]. Variations in Wnt signaling beyond β-catenin dependent/independent pathways also exist, further diversifying the biological functions of this essential cellular communication network [1,12,13,14,15].

The ten Frizzled proteins are regarded as the principle Wnt receptors and are instrumental for the transduction of all known Wnt signaling events, whereas the Wnt co-receptors LRP5/6 appear to be essential only for β-catenin-dependent as well as Wnt/STOP signaling [12,16,17,18]. With such a large repertoire of ligand–receptor interactions controlling Wnt signaling, a systematic approach to accurately quantify them under native conditions is important, and we have made considerable progress in this endeavor over the past few years [19,20,21,22]. Here, the term native condition corresponds to the study of full-length proteins in live cells using real-time measurements. Indeed, we have demonstrated that eGFP-Wnt3a conditioned medium (CM), when applied to HEK293TA cells overexpressing HiBiT-FZD_1–10_ constructs, allows for the comparative quantification of binding affinities and kinetics [20]. One downside of this study, however, was the use of HEK293A cells overexpressing high levels of HiBiT-FZDs, which underrepresents the true binding affinities [19,22]. Here, we report on our ongoing efforts to measure a wider range of Wnt-FZD interactions by NanoBiT/BRET-based assays using physiologically relevant receptor expression levels of full-length proteins. We have built on our ability to prepare full-length fluorescently tagged Wnt proteins as well as full-length HiBiT-tagged Frizzleds for real-time analysis in living cells. We show that a wide range of HiBiT-tagged FZD proteins, when expressed at low levels in living cells, provides a reliable comparison of binding characteristics for up to three different Wnt signaling proteins, Wnt3a, Wnt5a and Wnt16. Additionally, we demonstrate that the affinity of Wnt/FZD interactions is regulated by LRP6 in a manner that correlates with the latter’s signaling ability, which may provide some mechanistic insight of how Wnt signalosomes are formed.

## 2. Materials and Methods

### 2.1. Cell Culture and Ligands

Human embryonic kidney 293T (HEK293T) cells (CLS catalog number 300189), ΔFZD_1–10_ HEK293 cells [21], U-2 OS cells (ATCC^®^ HTB-96^TM^) and U-2 OS cell lines with a stable integration of mouse HiBiT-FZD_1,2,4,5,7,8,9,10_-mCherry (generated in this project) were cultured in Dulbecco’s modified Eagle medium (Gibco, Thermo Fisher, Waltham, MA, USA) supplemented with 10% fetal bovine serum (FBS, Thermo Fisher) and 1% penicillin–streptomycin (P/S; Gibco; Thermo Fisher, Waltham, MA, USA). NCI-H1703 cells with a stable integration of mouse HiBiT-FZD_1,4,5,8,10_-mCherry (generated in this project) were cultured in Roswell Park Memorial Institute (RPMI) 1640 medium (Gibco, Thermo Fisher, Waltham, MA, USA) supplemented with 10% fetal bovine serum, 1 mM sodium pyruvate (Gibco, Thermo Fisher, Waltham, MA, USA) and 1% P/S. HTC116 cells with a stable integration of mouse HiBiT-FZD1-mCherry (generated in this project) were cultured in McCoy’s 5a modified medium (Gibco, Thermo Fisher, Waltham, MA, USA) supplemented with 10% fetal bovine serum (FBS, Thermo Fisher, Waltham, MA, USA) and 1% penicillin–streptomycin (P/S; Gibco; Thermo Fisher, Waltham, MA, USA). All cell lines were maintained at 37 °C and in 5% CO_2_. Expi293^TM^ suspension cells (Thermo Fisher, A14527, Waltham, MA, USA) were cultured in Expi293^TM^ expression medium (Thermo Fisher, Waltham, MA, USA) at 37 °C and in 8% CO_2_ with 125 rpm orbital shaking in a New Brunswick S_4_1iCO_2_ shaking incubator (Eppendorf, Wesseling-Berzdorf, Germany). Cell densities and viability were determined using a Countess II automated cell counter (Life Technologies, Darmstadt, Germany).

For the preparation of eGFP-Wnt3a, eGFP-Wnt5a and eGFP-Wnt16 CM, Expi293^TM^ suspension cells growing in Expi293^TM^ expression medium (60 mL, 2.5 × 10^6^ cells/mL) were transfected with 10 µg of either pCS2+-eGFP-Wnt3a, pCS2+-eGFP-Wnt5a or pCS2+-eGFP-Wnt16, together with 50 µg of pCMV-His-Afamin using ScreenFect^®^ UP-293 (ScreenFect GmbH, Eggenstein-Leopoldshafen, Germany) according to the manufacturer’s instructions. eGFP-Wnt CM samples were collected 96 h post-transfection. The corresponding control CM was generated from cells transfected with pCS2+ plasmid. The CM was 5-fold-concentrated (Vivaspin turbo 30,000-molecular-weight-cutoff ultra filters—Satorius AG, Göttingen, Germany) followed by exchange to the desired cell culture medium using Sephadex G-25 PD10 desalting columns (GE Healthcare Bio-Science, München, Germany). The final concentration and stability of eGFP-Wnt-3a, eGFP-Wnt5a and eGFP-Wnt16 in the CM samples were determined using Western blotting and ELISA (GFP ELISA^®^ kit, Abcam, ab171581, Cambridge, UK).

### 2.2. Generation of eGFP-Wnt CM in HEK293T Cells

A total of 1.2 × 10^6^ HEK293T cells in 6 wells were co-transfected with 0.7 µg of pCMV3-His-hAfamin and 1 µg of pCS2+ together with 0.3 µg of pCS2+ eGFP-Wnt fusion construct for Wnt1, Wnt2b, Wnt3a, Wnt5a, Wnt6, Wnt7a, Wnt8a, Wnt9a, Wnt10a, Wnt10b, Wnt11 and Wnt16 or co-transfected with 1.7 µg of pCS2+ and 0.3 µg of pCS2+ eGFP-Wnt fusion construct using ScreenFect^®^ A (ScreenFect GmbH, Eggenstein-Leopoldshafen, Germany) according to the manufacturer’s 1-step protocol. A total of 72 h post-transfection, the medium was discarded, and 2 mL of fresh medium was added to the cells and incubated for a further 72 h before harvest. Conditioned medium (CM) was harvested and centrifuged for 5 min at 1500× *g* to pellet possible dead cells. A total of 1.5 mL of CM was mixed with 75 µL of 20% Triton-X 100 to a final concentration of 1% Triton-X 100. Cibacron blue 3G coupled to Sepharose 6 Fast Flow (Blue-Sepharose 6 Fast Flow, GE Healthcare, München, Germany) beads were washed three times with Blue-Sepharose buffer (BS-buffer) (150 mM KCl, 50 mM Tris-HCl, pH 7.5, 1% Triton X-100) containing Complete^®^ protease inhibitor mixture (Roche, Mannheim, Germany). Washed beads were added to the eGFP-Wnt CM and rotated overnight at 4 °C. The next day, the beads were washed three times in BS-buffer, Laemmli sample buffer was added, and the samples were heat-denaturated. SDS-PAGE/Western blotting was performed, and secreted eGFP-Wnt proteins were detected using anti-GFP antibody (Santa Cruze Biotechnology, sc-9996, Heidelberg, Germany).

### 2.3. Plasmids

NFAT and AP-1 luciferase reporter were generated by replacing TCF/LEF binding sites of M50 Super 8xTOPFLASH (a gift from Randall Moon, Addgene plasmid 12456, Watertown, MA, USA) by NFAT or AP-1 response elements. For NFAT, the binding site from the human interleukin 2 (IL-2) gene (Mattila et al., 1990) was ordered as complementary DNA oligos flanked by KpnI and XhoI sites (NFAT binding site, 5′-AAC TCG AGC GCC TTC TGT ATG AAA CAG TTT TTC CTC CGG TAC CAA A-3′). The oligos were annealed and inserted between the KpnI and XhoI restriction sites of the M50 plasmid. For AP-1 luciferase reporter, the AP-1 response element of three repeats of two alternative AP-1 binding sites [23] were ordered as oligos flanked by KpnI and NheI restriction sites (AP-1 binding site, 5′-AAG GTA CCT GAG TCA GTG ACT CAG TGA GTC AGT GAC TCA GTG AGT CAG TGA CTC AGC TCG AGA AA-3′). The oligos were annealed and inserted between the KpnI and NheI restriction sites of the M50 plasmid. pCMV3-His-hAfamin was obtained from Sino Biological (HG13231-CH), pCS2+ eGFP-Wnt-3a (Wesslowski et al., 2020). N-terminal eGFP-(GGSG)-Wnt fusion proteins for Wnt1, Wnt2b, Wnt5a, Wnt6, Wnt7a, Wnt8a, Wnt9a, Wnt10a, Wnt10b, Wnt11 and Wnt16 were generated by inserting the synthetic human codon-optimized ORF of eGFP-Wnt DNA fragment (Invitrogen GeneArt Strings DNA Frgaments, Thermo Fisher Scientific, Darmstadt, Germany) into pCS2+ expression vector using BamHI and XbaI restriction sites. To generate pEF1α HiBiT-FZD_1,2,4,6,7,8,9,10_-mCherry-IRES-NEO, FZD_1,2,4,6,7,8,9,10_-mCherry ORFs without signal peptides were PCR-amplified from the corresponding pmCherry-FZD-mCherry plasmids [21] flanked by XbaI and NotI restriction sites. 5-HT_3_A signal peptide followed upstream by HiBiT sequence [21] and flanked by XhoI and XbaI restriction sites was ordered as complementary DNA oligos (5′-TCG AGG CCA CCA TGC GGC TCT GCA TCC CGC AGG TGC TGT TGG CCT TGT TCC TTT CCA TGC TGA CAG GGC CGG GAG AAG GCA GCC GGG TGA GCG GCT GGC GGC TGT TCA AGA AGA TTA GCT-3′ (forward) and 5′-CTA GAG CTA ATC TTC TTG AAC AGC CGC CAG CCG CTC ACC CGG CTG CCT TCT CCC GGC CCT GTC AGC ATG GAA AGG AAC AAG GCC AAC AGC ACC TGC GGG ATG CAG AGC CGC ATG GTG GCC-3′ (reverse)). The oligos were annealed and, together with the amplified FZD1,2,4,6,7,8,9,10-mCherry ORFs, inserted between the XhoI and NotI restriction site of pEF1α -IRES-NEO (a gift from Thomas Zwaka, Addgene plasmid 28019, Watertown, MA, USA).

### 2.4. Generation of Cell Lines Stably Expressing HiBiT-FZD-mCherry

NCI-1703, HTC116 or U-2 OS cells were transfected with 2 µg of pEF1a-HiBiT-FZD-mCherry-IRES-Neo in 6-well plates using ScreenFect^®^ A (ScreenFect GmbH, Eggenstein-Leopoldshafen, Germany) according to the manufacturer’s 1-step protocol, followed by a medium exchange 24 h post-transfection. A total of 48 h post-transfection, the medium was exchanged, and after further 48 h, the cells were transferred to 10 cm^2^ dishes, and selection was initiated by adding 1 mg/mL G418 (Sigma-Aldrich, Taufkirchen, Germany). After 7 days of selection, cells were FACS-sorted according to cell surface receptor expression levels.

### 2.5. Fluorescence-Activated Cell Sorting (FACS)

FACS was performed with a FACSAria^TM^ Flow Cytometer (BD Biosciences, Heidelberg, Germany). In order to sort U-2 OS for stable integration of the HiBiT-FZD-mCherry gene with cell-surface-located HiBiT-FZD-mCherry, cells were detached using 5 mM EDTA (Roth, Karlsruhe, Germany) in PBS^-/-^ (Gibco, Thermo Fisher, Waltham, MA, USA) and collected in DMEM (Gibco) supplemented with 10% FBS (Gibco). A total amount of approximately 1 × 10^7^ cells was used for sorting. Cells were resuspended in 1 mL of ice-cold FACS buffer (2% FBS; 2 mM EDTA; PBS^-/-^). Surface HiBiT-FZD-Cherry was labeled with 1 µg/mL anti-HiBiT antibody (#N7200 Promega, Walldorf, Germany) in FACS buffer for 45 min on ice. Cells were washed two times with 5 mL of ice-cold FACS buffer and labeled with 2.5 µg/mL anti-mouse Alexa 488 (A21204, Invitrogen, Thermo Fisher, Waltham, MA, USA) for 30 min on ice. Cells were washed two times with 5 mL of FACS buffer and resuspended in 4 mL of FACS buffer for sorting. Gating for low, medium and high cell surface protein signals was carried out using the BD FACSDiva^TM^ v9.0 software, and cells were single-cell-sorted in 96 well plates. Five weeks after sorting, clonal lines were re-analyzed with FACS for their cell surface expression levels and verified via FACS analysis.

### 2.6. Reporter Gene Assay

To test the biological activity of eGFP-Wnt-3a, eGFP-Wnt-5a and eGFP-Wnt-16, 5.5 × 10^4^ HEK293T cells cultured in 96-well plates were transfected with 20 ng of TCF firefly luciferase (TOP-FLASH), 2 ng of CMV Renilla luciferase and 78 ng of pCS2+ empty plasmid using ScreenFect^®^ A according to the manufacturer’s 1-step protocol (ScreenFect GmbH, Eggenstein-Leopoldshafen, Germany). A total of 24 h post-transfection, the medium was replaced by control, eGFP-Wnt-3a CM, eGFP-Wnt-5a CM or eGFP-Wnt-16 CM, and cells were incubated for another 24 h before harvesting cell lysates in 1 × Passive Lysis Buffer (Promega, Walldorf, Germany). Experiments were performed at least 3 times, unless indicated otherwise in the figure legends. Error bars shown are standard deviations from the mean (±S.D.) of the indicated number of n = 4 independent biological samples within an experiment.

### 2.7. WB Analysis

eGFP-Wnt-3a, eGFP-Wnt-5a and eGFP-Wnt-16 were mixed with Laemmli sample buffer and heat-denatured. Samples were separated by SDS-PAGE before being transferred to a PVDF membrane using a Bio-Rad Transblot-Turbo system (Bio-Rad, Feldkirchen, Germany). Membranes were blocked at room temperature for 1 h in 5% BSA–TBST blocking buffer (5% BSA, 137 mM NaCl, 2.7 mM KCl, 19 mM Tris base [pH 7.4], 0.1% Tween-20) and transferred to a BioLane HTI automated Western blotting processor for antibody incubation and washing steps. The following antibodies were used: anti anti-GFP (ab1828, 1:2000, Abcam, Cambridge, UK) and HRP-conjugated anti-rabbit or anti-mouse secondary antibodies (Dako, Agilent, Waldbronn, Germany). For semiquantitative detection of protein bands, the membranes were incubated with ECL Prime (GE-Healthcare Bio-science, München, Germany) and imaged using a ChemiDocTM touch imaging system (Bio-Rad, Feldkirchen, Germany).

### 2.8. NanoBiT/BRET Binding Assay

U-2 OS cells stably expressing HiBiT-FZD-mCherry were seeded as 6500 cells/well onto a poly-D-lysine (Gibco, Thermo Fisher, Schwerte, Germany)-coated white 96-well cell culture plate with a clear flat bottom (VWR part of avantor). Twenty-four hours later, the cells were washed once with 200 µL of non-phenol red DMEM (Gibco, Thermo Fisher) supplemented with 5% FBS and 10 mM HEPES (Gibco, Thermo Fisher, Schwerte, Germany). The cells were preincubated with 50 µL of non-phenol red DMEM supplemented with Vivazine (1:100 dilution; Promega), LgBiT (1:100 dilution, Promega, Walldorf, Germany), 5% FBS and 10 mM HEPES for 1 h at 37 °C without CO_2_. Subsequently, 50 µL of four different concentrations of eGFP-Wnt-3a (1.68, 3, 7.5 and 15.2 nM), eGFP-Wnt-5a (2.2, 4.3, 8.7 and 17.3 nM) and eGFP-Wnt-16 (3, 6, 12 and 24 nM) conditioned medium or control conditioned medium supplemented with Vivazine (1:150 dilution, Promega, Walldorf, Germany), 5% FBS and 10 mM HEPES were added, and the BRET signal was measured every 80 s for 300 min at 37 °C.

For the NanoBiT/BRET assay with stable HiBiT-FZD_1_-mCherry U-2 OS cells co-expressing B3GNT2 and LRP6, a total of 10,000 cells were seeded in poly-d-lysine (Gibco, Thermo Fisher, Schwerte, Germany)-coated white 96-well cell culture plates with a clear bottom. After 24 h, the cells were transfected with lacZ (control up to 100 ng), B3GnT2 (1 ng), LRP6 (20 ng) and MESD (5 ng) using Lipofectamine 3000 Transfection Reagent (Thermo Fisher, Schwerte, Germany). The next day, the cells were washed once with 200 μL of non-phenol red DMEM supplemented with 10 mM HEPES and 5% FBS. The cells were preincubated with 50 μL of a mix of Vivazine (1:50 dilution) and LgBiT (1:100 dilution) in complete non-phenol red DMEM supplemented with 10 mM HEPES for 1 h at 37 °C without CO_2_. Subsequently, 50 μL of four different concentrations of eGFP-Wnt3a (1.68, 3, 7.5 and 15.2 nM), eGFP-Wnt5a (2.2, 4.3, 8.7 and 17.3 nM) or eGFP-Wnt16 (3, 6, 12 and 24 nM) CM supplemented with 5% FBS and 10 mm HEPES were added, and subsequently, the eGFP fluorescence was measured (excitation, 470–15 nm; emission, 515–20 nm). Prior to BRET measurement, a white foil was stuck to the underside of the plates in order to increase the luminescence detection. BRET acceptor (bandpass filter, 535–30 nm) and BRET donor (bandpass filter, 450–80 nm) emission signals were measured every 80 s for 300 min at 37 °C using a CLARIOstar microplate reader (BMG Labtech, Ortenberg, Germany). Data were analyzed using GraphPad Prism 8 (San Diego, CA, USA).

### 2.9. Estimation of Receptor Density

Stable U-2 OS HiBiT-FZD_1_-mCherry cells were seeded in a white 96-well plate with a transparent bottom, and 48 h later, the measurement was started. To prepare a standard curve, 15 samples of HiBiT control protein (Promega, #N3010, Walldorf, Germany) were prepared, ranging from 1 fM to 100 mM, in non-phenol red DMEM supplemented with 5% FBS and 10 mM HEPES, and 90 µL/well each was distributed as duplicates in white 96-well plates with transparent bottoms. Cells were washed once with 200 µL of non-phenol red DMEM and covered with 90 µL of non-phenol red DMEM supplemented with 5% FBS and 10 mM HEPES. The reaction mix was prepared by supplementing non-phenol red DMEM with LgBiT (1:20 Promega, Walldorf, Germany) and Furimazine (1:10, Promega, Walldorf, Germany). A total of 10 µL of the reaction mix was added to each well, carefully mixed and incubated for 10 min at 37 °C without CO_2_. A white sticker was stuck to the bottom of the plate, and the luminescence emitted by the complemented NanoBiT luciferase was measured (460–500 nm, 200 ms integration time). Ater measurement, cells were detached and counted using a Countess II automated cell counter (Life Technologies, Darmstadt, Germany). Luminescence/cell was calculated, and, together with the standard curve and the molecular weight of the HiBiT control protein, the receptor number/cell was calculated.

### 2.10. Data Analysis and Statistics

All data analyses presented were performed at least three times in duplicates. Data points on the binding curves represent mean ± SEM. Kinetic binding was analyzed using the association model with two or more hot ligand concentrations in GraphPad Prism 8 (San Diego, CA, USA). Binding affinity values (Kd) are presented as a best-fit Kd with SEM. The difference between the k_d_ values obtained within each experimental assay were compared for statistical significance using a Kruskal–Wallis rank sum test, which is shown in Appendix A, along with a reference to the software program R (version 4.5), which was used.

### 2.11. Confocal Laser Scanning Microscopy

For the microscopy analysis of eGFP-Wnt fusion proteins binding to Frizzled receptors, NCI-1703 cells with a stable integration of the HiBiT-FZD_4,5,8,10_-mCherry gene were seeded in µ-Slide 18-well chambers (Ibidi, catalog no. 81816, Gräfelfing, Germany) so that they were 80% confluent for microscopy after 2 days. Prior to imaging, the individual eGFP-Wnt CM for Wnt1, Wnt2b, Wnt3a, Wnt5a, Wnt6, Wnt7a, Wnz8a, Wnt9a, Wnt10a, Wnt10b, Wnt11 and Wnt16 were added to the cells and incubated for 1 h. For microscopy, cell medium was exchanged with a mix of 50% FluoroBrite DMEM (Gibco, Thermo Fisher, Schwerte, Germany) and 50% non-phenol red RPMI (Gibco, Thermo Fisher) supplemented with 10% FBS and 10 mM HEPES (Gibco, Thermo Fischer, Schwerte, Germany). For microscopy analysis of eGFP-Wnt3a binding to Frizzled receptors in U-2 OS cells, U-2 OS cells with a stable integration of HiBiT-FZD_1,2,3,4,5,6,7,8,9,10_-mCherry were seeded in µ-Slide 18-well chambers (Ibidi, catalog no. 81816, Gräfelfing, Germany) so that they were 80% confluent for microscopy after 2 days. Prior to imaging, eGFP-Wnt3a CM was added to the cells and incubated for 3 h. For microscopy, cell medium was exchanged with FluoroBrite DMEM (Gibco, Thermo Fisher, Schwerte, Germany) supplemented with 10% FBS and 10 mM HEPES (Gibco, Thermo Fischer, Schwerte, Germany). Confocal images were taken using a Zeiss LSM 800 microscope (Zeiss, Jena, Germany) fitted with a 40×/1.2 oil differential interference contrast (UV) VIS-IR Plan-Apochromat objective (Zeiss, Jena, Germany) and a GaAsP-PMT detector. Images were analyzed using Fiji [24].

## 3. Results

### 3.1. Text Description of Results

#### 3.1.1. Fluorescent Tagging of Wnt5a and Wnt16 Preserves Secretion and Signaling

Wnts are secreted lipoproteins that are difficult to purify in an active state, and their tagging is challenging, especially larger fluorescent moieties. Wnt3a and Wnt5a were the first to be purified from conditioned medium [11,25], and more recently, Wnt3a has been successfully fluorescently tagged with only partial loss of signaling activity [21,26].

To expand on our analysis of Wnt/FZD interactions using NanoBiT/BRET [19,20], we added an N-terminal eGFP to eleven Wnt proteins (Wnt1, Wnt2b, Wnt5, Wnt6, Wnt7a, Wnt8a, Wnt9a, Wnt10a, Wnt10b, Wnt11 and Wnt16), following the same principle as we described previously for Wnt3a [21]. Barring eGFP-Wnt11, all fusion proteins were secreted and detected in the medium (Appendix A, see the Materials and Methods section for details). Co-expression of the Wnt-binding protein Afamin [27] further increased the amounts detected in the medium for eGFP-Wnt1, -2b, -3a, -7a, -10a, -16 and especially for eGFP-Wnt6, -9a and -10b. Furthermore, Afamin co-expression enabled secretion of Wnt11 into the medium (Appendix A). We therefore routinely included Afamin co-expression for the preparation of condition medium (CM) containing eGFP-Wnt.

Using confocal microscopy, we next tested the ability of the fluorescently tagged Wnt proteins to associate with NCI-H1703 cells stably expressing HiBiT-FZD_5_ (Figure 1A).

Upon addition of CM, all Wnts, apart from Wnt11, showed specific membrane association to varying degrees, with Wnt3a and Wnt5a presenting the strongest binding. Interestingly, only Wnt3a, Wnt5a and, to a lesser extent, Wnt16 presented uniform membrane association, with Wnt1, -2b, -6, -7a, -8a, -9a, -10a and -10b displaying only partial binding to cell membranes that appeared asymmetric in nature (see arrow in Figure 1A and Appendix A). Indeed, eGFP-Wnt3a, -5a and -16 displayed the most consistent association with a variety of FZD-expressing cells (Appendix A), and we choose these three ligands for a first comparative study of Wnt-FZD interactions. Reporter assays for β-catenin-dependent (TOPFLASH) and β-catenin-independent (ATF2, NFAT and AP1) Wnt signaling confirmed that eGFP-Wnt3a, eGFP-5a and eGFP-16 retained functional activity (Figure 1B), and CM samples for these were prepared using Expi293 suspension cells (see the Materials and Methods section for details). Western blot analysis of the highest protein concentrations used in the NanoBiT/BRET assay confirmed that there was no obvious degradation of the fusion proteins, and we obtained precise concentrations for each: eGFP-Wnt3a 15.2 nM; eGFP-Wnt5a 17.3 nM; eGFP-Wnt16 24 nM (Figure 1C).

#### 3.1.2. Generation of HiBiT-FZD Stable Cell Lines

We have previously shown that transient transfection of HiBiT-FZD in HEK293T cells allows for the reliable measurement of eGFP-Wnt3a binding [20]. To obtain a more physiologically relevant cell-culture-based system for measuring Wnt-FZD interactions, we used HCT116, NCI-H1703 and U-2 OS cells to generate stable cell lines with low-level expression of HiBiT-FZD1. We were unable to generate stable, low-HiBiT-FZD1-expressing NCI-1703 cell lines, but the larger U-2 OS cells and smaller HCT116 cells were more amenable (Appendix A). Although the average numbers of receptors per cell for the U-2 OS and HCT116 cells were similar, receptor density was significantly lower on U-2 OS due to their larger size (5- to 10-fold larger), and, correspondingly, a weaker fluorescence signal at the cell surface was detected in the U-2 OS cells (Appendix A). Next, to study the effect of receptor levels on eGFP-Wnt3a binding using NanoBiT/BRET assays, three different stable U-2 OS cell populations with varying receptor densities were prepared (Figure 2A,B).

Stable cell lines expressing lower HiBiT-FZD1 levels displayed a higher binding affinity for eGFP-Wnt3a, which is in line with our previous findings [22] (Figure 2C). This is likely due to a more physiological environment for ligand–receptor binding, minimizing out-titration of endogenous cellular components that play roles in Wnt receptor engagement. Expectedly, a higher standard deviation of measurements was seen in U-2 OS cells expressing lower receptor levels (Figure 2C, compare error bars in binding curves). For all subsequent analyses, we employed FACS-sorted stable U-2 OS cell lines expressing low/medium levels of HiBiT-FZD_1–10_-mCherry, corresponding to receptor densities of between 500 (low) and 2000 (medium) per cell (Figure 2A, Appendix A). The signaling activity of HiBiT-FZD_1–10_-mCherry proteins was tested using TOPFLASH Wnt reporter assays after expressing the corresponding constructs in ∆FZD_1–10_^GFP-free^ HEK293 cells [21]. No signaling activity was detected for FZD3 or FZD6 fusion proteins, which is in line with their preference for non-canonical Wnt pathways (Appendix A). FZD8 displayed only weak signaling, whereas all others transduced Wnt/β-catenin signaling to a similar degree (Appendix A). Interestingly, although HiBiT-FZD9-mCherry showed strong signaling activity in TOPFLASH assays, eGFP-Wnt3a binding was undetectable in HiBiT-FZD9-mCherry U-2 OS cells (Appendix A).

#### 3.1.3. NanoBiT/BRET Analysis of Wnt3a, Wnt5a and Wnt16 to HiBiT-FZD_1–10_

For real-time NanoBiT/BRET analysis, which is schematically illustrated in Figure 3A, we FACS-sorted a wide range of U-2 OS cell lines expressing low levels of stably integrated HiBiT-FZD_1–10_, as described above (Figure 3, Appendix A).

The relative levels of HiBiT-FZD’s at the cell surface were estimated from the NanoBiT luminescence generated upon addition of LgBiT and substrate to the cells. Generally, cell surface receptor levels did not vary by more than 2-fold between experiments; however, FZD8 and FZD9 showed lower levels, and FZD10 showed higher levels (Figure 3B). Four concentrations of EGFP-Wnt3a, -5a and -16 were used to generate the association curves needed for calculating the kinetic binding affinities. Note that, for some Wnt-FZD combinations, accurate association curves could not be fitted when using either the highest or lowest concentration, and one or the other had to be omitted to allow for the precise calculation of the binding affinity (Appendix A). Of the 30 possible Wnt-FZD combinations for Wnt3a, Wnt5a and Wnt16, binding affinities for 19 pairs could be measured using stable U-2 OS cell lines (Figure 3C). No concentration-dependent increase in the BRET ratio was detected for U-2 OS cells expressing either HiBiT-FZD3 or HiBiT-FZD6, in line with their inability to transduce Wnt/β-catenin signaling (Figure 3, Appendix A). U-2 OS cells expressing HiBiT-FZD9 also failed to show detectable association curves, despite the robust transduction of Wnt/β-catenin signaling (Figure 3, Appendix A). Indeed, this would fit with the lack of eGFP-Wnt3a binding seen for HiBiT-FZD9 expressing U-2 OS cells (Appendix A).

For Wnt3a association to the HiBiT-FZD cell lines, the binding affinity, from highest to lowest, was as follows: FZD_7_ > FZD_4_ > FZD_2_ > FZD_5_ > FZD_8_ > FZD_1_ > FZD_10_. This is broadly in agreement with our previous work using HEK293 cells transiently transfected with the same HiBiT-FZD constructs, which express at far higher levels at the cell membrane [20]. Like this study, the BRET ratio binding curves for FZD_3_, FZD_6_ and FZD_9_ were mostly undetectable or very low in this previous work [20]. Generally, the k_d_ values calculated here using HiBiT-FZD’s expressed at a low level in U-2 OS cells were around 10-fold higher compared to the affinities we calculated previously using transiently transfected HEK293 cells. Although the order of increasing binding affinities was broadly similar between the two studies, one notable exception was FZD_10_, which displayed significantly lower affinity here (12.5 nM for stable U-2 OS cells compared to 4.3 nM in transiently transfected HEK).

For Wnt5a, the FZD binding affinities were, on average, 7-fold lower compared to Wnt3a, although this reduction was greater/less for FZD_5_/FZD_8_, respectively (Figure 3C,D). Similar to eGFP-Wnt3a binding, the strongest association with eGFP-Wnt5a was seen for FZD_7_, followed by FZD_4_ and FZD_2_ (Figure 3D). Indeed, the overall order of binding affinities to FZD’s for Wnt3a and Wnt5a were similar (Wnt3a: FZD_7_ > FZD_4_ > FZD_2_ > FZD_5_ > FZD_8_ > FZD_1_; Wnt5a: FZD_7_ > FZD_4_ > FZD_2_ > FZD_8_ > FZD_1_ > FZD_5_).

Wnt16 had the weakest binding affinities (higher k_d_’s) overall to the HiBiT-FZD cell lines, with FZD_8_, rather than FZD_7_, showing the strongest (Figure 3C,D). In contrast to Wnt3a, neither Wnt16 nor Wnt5a generated strong enough binding curves to allow for the calculation of the binding affinities to HiBiT-FZD_10_.

#### 3.1.4. LRP6 Influences the Binding Affinity Between Wnt3a and HiBiT-FZD_1_

Our results demonstrate that NanoBiT/BRET assays can reliably quantify the binding affinities of a wide variety of Wnt/FZD interactions at the cell surface. Wnt proteins, however, additionally interact with the co-receptor LRP6, forming a trimeric Wnt-FZD-LRP6 complex to transduce Wnt/β-catenin signaling. We therefore used NanoBiT/BRET to study the influence of LRP6 on Wnt/FZD interactions (Figure 4A).

To this end, we transfected stable U-2 OS HiBit-FZD_1_ cells with either LRP6 or a control pDNA (LacZ) and measured the binding of eGFP-Wnt3a. Compared to FACS-sorted U-2 OS cells stably expressing HiBiT-FZD_1_, their transiently transfected control counterparts displayed slightly lower Wnt3a-FZD_1_ binding, with affinities of 3.5 nM and 9.6 nM, respectively (Figure 4B, upper left graph, and Figure 3D). Somewhat surprisingly, overexpression of LRP6 resulted in a 3-fold decrease in Wnt3a-FZD_1_ binding affinity (9.6 nM to 27.9 nM) (Figure 4B, compare upper two graphs). We also tested the LRP6 modifier, B3GnT2, which we recently showed can activate Wnt/β-catenin signaling by modifying multiple N-glycans on its extracellular domain [28]. Of note, co-transfection of LRP6 with B3GnT2 resulted in a robust (5-fold) increase in Wnt-FZD binding affinity, from 27.9 nM to 6.1 nM (Figure 4B, compare graphs on right). A similar, albeit less robust, increase in Wnt-FZD binding affinity was seen upon transfection of B3GnT2 alone (from 9.6 nM to 3.4 nM) (Figure 4B, compare graphs on left), suggesting that B3GnT2 may act on endogenous LRP6 and/or FZD to regulate Wnt-FZD association. The amount of HiBiT-FZD_1_ expressed at the cell surface of the U-2 OS cells used in these experiments was similar for all conditions (Figure 4C). Taken together, these results suggest that NanoBiT/BRET analysis of Wnt-FZD interactions is a valuable method for studying the signaling capability of the trimeric Wnt-FZD-LRP6 complex.

## 4. Discussion

We continue to develop NanoBiT/BRET as a standard method for real-time quantification of ligand–receptor interactions within the Wnt pathway using full-length proteins in living cells. We have now generated a wide range of cell lines that stably express low levels of HiBiT-FZD_1–10_, and we have demonstrated that they provide a more accurate system for analysis of Wnt-FZD binding. We have used these cell lines for NanoBiT/BRET analysis together with three different eGFP-Wnt proteins, and we have generated a more comprehensive data set of comparative Wnt-FZD binding affinities. This study was limited to Wnt3a, Wnt5a and Wnt16 because they displayed the clearest binding to FZD. Nevertheless, ongoing efforts to expand the availability and use of different fluorescently tagged Wnt’s continue. One encouraging aspect is the apparent ease with which eGFP-tagged Wnt proteins are secreted from cells and obtainable as soluble proteins within the culture medium. The Wnt-binding protein Afamin [27] clearly helps the accumulation of eGFP-Wnt proteins in the extracellular medium and, thus, simplifies the preparation of Wnt conditioned medium; nevertheless, its effectiveness varies between Wnts. Wnt11 appears to be entirely dependent on Afamin for secretion; others, such as Wnt6, are barely detected in the medium in its absence, whereas still others, such as Wnt5a, Wnt8a and Wnt10a, appear less dependent. Different mechanisms are reported to account for delivery of Wnt to the extracellular environment [29,30,31,32], and specific Wnts may have mechanistic preferences, perhaps explaining the differences in how Afamin affected our results.

It is informative to compare the work presented here using the stable U-2 OS cell lines with our previous study using transient transfection of HEK293 cells, which produced significantly higher receptor levels [20]. On average, we saw a near 10-fold increase in Wnt-FZD binding affinities using the stable U-2 OS cell lines expressing low HiBiT-FZD levels. This fits well with our hypothesis that accurate and precise quantification of ligand–receptor interactions is best achieved when signaling proteins interact with receptors at the cell surface under physiologically relevant conditions. Fortunately, there appears to be a range of such physiological relevant “low” expression levels because we have previously shown that cell lines stably expressing tagged receptors exhibit similar binding affinities when compared to cells with endogenously tagged receptors, despite having significantly higher cell surface levels [22]. Transiently transfected cells, on the other hand, which express far higher receptor levels, have significantly lower binding affinities. Thus, we can confidently use stable cell lines with relatively low levels of tagged receptors (between 500 and 2000 receptors per cell according to our calculations) and do not necessarily require CRISPR-mediated endogenous tagging.

In contrast to the strong differences in Wnt3a-FZD binding affinities between transiently transfected HEK293 cells and U-2 OS stable cell lines, the relative order of binding affinities between the two studies remained similar. Our findings are also in relatively good agreement with a previous study using different Wnt/FZD combinations to rescue Wnt/β-catenin signaling in FZD^-/-^ cells [33], where FZD_2_, FZD_4_, FZD_5_ and FZD_7_ showed the strongest activities. One exception, however, is FZD_10_, which showed a relatively strong ability to rescue [33]; however, it presented relatively low BRET ratio values in this study, in spite of showing one of the highest cell surface expression levels (Figure 3B). The reason for this discrepancy is unclear, but different cellular contexts are the most likely possibility, as different cell lines will have different expression profiles of Wnt receptor/co-receptors. Another interesting discrepancy highlighted by our study was the strong signaling activity of FZD_9_ in TOPFLASH assays, although there was an almost complete lack of ability to associate with Wnt at the membrane. One potential explanation for the discrepancy is a more rapid internalization of HiBiT-FZD_9_ compared to other FZD’s, leading to rapid removal from the cell surface upon engagement with Wnt. This may not influence the TOPFLASH assay but may negatively influence BRET measurement at the cell surface. Nevertheless, the previous study mentioned above also showed that FZD_9_ was unable to rescue signaling when re-introduced in FZD^-/-^ cells [33], perhaps indicating some unclear characteristics of this particular receptor, and future studies should provide answers.

In summary, the relative consistency between our studies using different cells indicates that NanoBiT/BRET binding should be suitable for a variety of cells; however, future studies using a greater variety of cell types is needed to confirm this.

Despite the markedly lower overall binding strengths to FZDs for Wnt5a compared to Wnt3a, one somewhat surprising finding was the similarity for Wnt3a and Wnt5a in their relative order of binding affinities to the different FZDs. This would point to the nature of binding between Wnt/FZD being mostly attributable to the more conserved elements of Wnts. Nevertheless, Wnt16 did show some striking differences, such as FZD_8_ being the strongest binding partner and FZD_4_ one of the weakest, which is in contrast to the binding profiles of either Wnt3a or Wnt5a.

In our experiments using HiBiT-FZD_1_-expressing cells with/without LRP6 co-expression, LRP6 appeared to regulate the binding strengths between Wnt and FZD within the Wnt-FZD-LRP6 complex. This may provide some additional insight with respect to the mechanisms governing Wnt signalosome formation. Evidence exists that one Wnt molecule can associate with more than one FZD-CRD unit [34]. It is tempting to hypothesize that LRP6 can influence the ability of Wnt to form different stoichiometries with FZD’s, and future work using tools such as NanoBiT/BRET-based ligand–receptor interactions should help provide answers. Additionally, our results demonstrate that enhanced glycosylation of the extracellular region of LRP6 by B3GnT2 also influences Wnt-FZD association. It is noteworthy that B3GnT2 extends polylactosamine chains close to potential Wnt binding sites on LRP6 and may therefore sterically alter Wnt-FZD interactions. This may indeed partially account for B3GnT2’s ability to enhance Wnt/LRP6 signaling [28].

Our strategic use of the NanoBiT/BRET methodology to accurately quantify the large repertoire of ligand–receptor interactions within the Wnt signaling pathway appears to be a valid approach. The generation of a wide range of cell lines expressing physiologically relevant levels of HiBiT-FZD has allowed us, for the first time, to assemble a more global comparative overview as well as accurate quantification of Wnt-FZD interactions on cell membranes.

## Figures and Tables

**Figure 1 cells-14-00810-f001:**
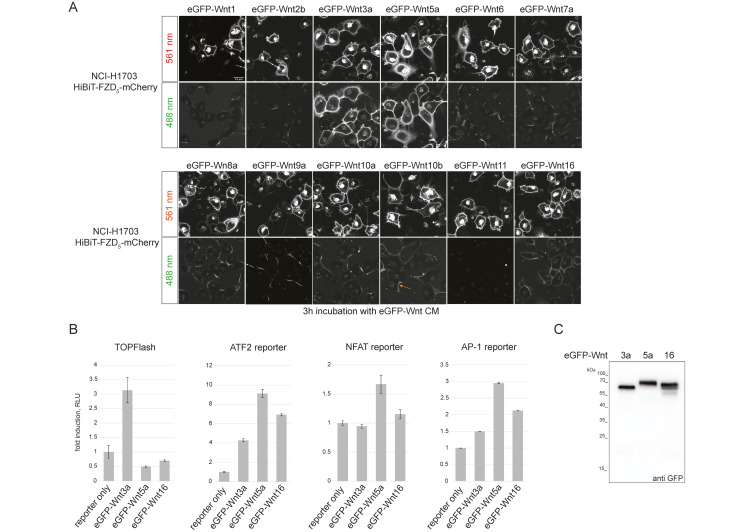
Generation and testing of eGFP-Wnt3a, -5a and -16 CM. (**A**) Laser scanning confocal microscopy images of NCI-H1703 cells with stable integration of mouse HiBiT-FZD5, incubated for 3 h with the indicated eGFP-Wnt CM derived from Expi293TM suspension cells. The arrow in the lower eGFP-Wnt10b panel points to an example of asymmetric binding observed for several Wnts (see Appendix A for higher-resolution images). (**B**) Reporter gene assay using HEK293T cells transfected with TOPFLASH reporter to measure the activation of the canonical Wnt pathway and ATF2, NFAT and AP1 luciferase reporter to detect β-catenin-independent Wnt pathway activation. Cells were treated with eGFP-Wnt3a, eGFP-Wnt5a and eGFP-Wnt16 CM for 24 h. Error bars represent means ± S.D. from 4 independent samples. Experiments were performed 3 times with similar results. (**C**) Western blot using anti GFP antibody, showing presence of soluble eGFP-Wnt3a, eGFP-Wnt5a and eGFP-Wnt16 in conditioned medium (CM) derived from Expi293TM suspension cells, representing the highest concentration used for the NanoBiT/BRET assay: 15.2 nM for eGFP-Wnt3a, 17.3 nM for eGFP-Wnt5a and 24 nM for eGFP-Wnt16.

**Figure 2 cells-14-00810-f002:**
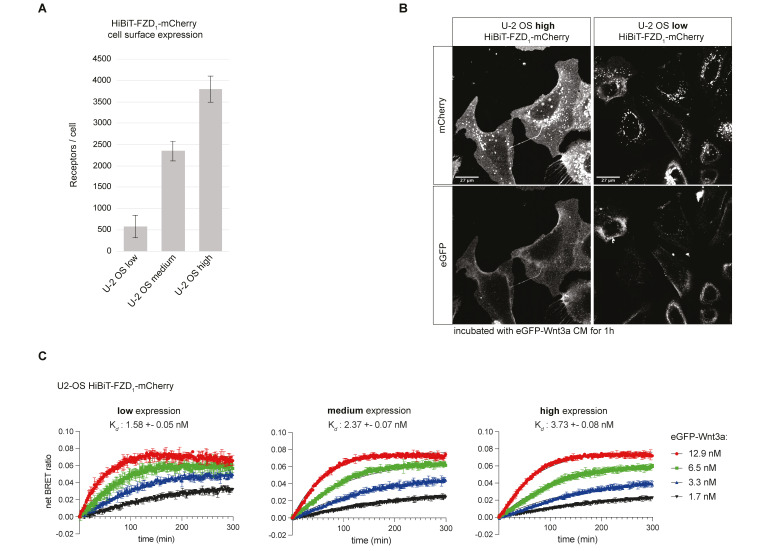
Comparison of Wnt-FZD binding in U-2 OS stable cell lines with different levels of HiBiT-FZD1-mCherry. (**A**) Indirect quantification of HiBiT-FZD1-mCherry at the cell surface of FACS-sorted U-2 OS stable cell lines expressing three different levels of HiBiT-FZD1-mCherry. The average number of cell-surface-located receptor molecules was estimated using commercially available HiBiT control protein as a reference. (**B**) Laser scanning confocal microscopy image of FACS-sorted U-2 OS cells with stable integration of a HiBiT-FZD1-mCherry gene, expressed at high and low levels. (**C**) Association kinetics of eGFP-Wnt3a binding to HiBiT-FZD1-mCherry were determined by NanoBiT/BRET in FACS-sorted U-2 OS stable cell lines over a 5 h time period using 1.68, 3.36, 6.48 and 12.96 nM of eGFP-Wnt3a, measured every 80 s. Raw data were fitted to the ‘two or more hot concentrations model’ and are presented as mean ± S.D from n = 3 individual experiments. Three cell lines with different membrane receptor numbers/cell were used and labeled as low, medium or high.

**Figure 3 cells-14-00810-f003:**
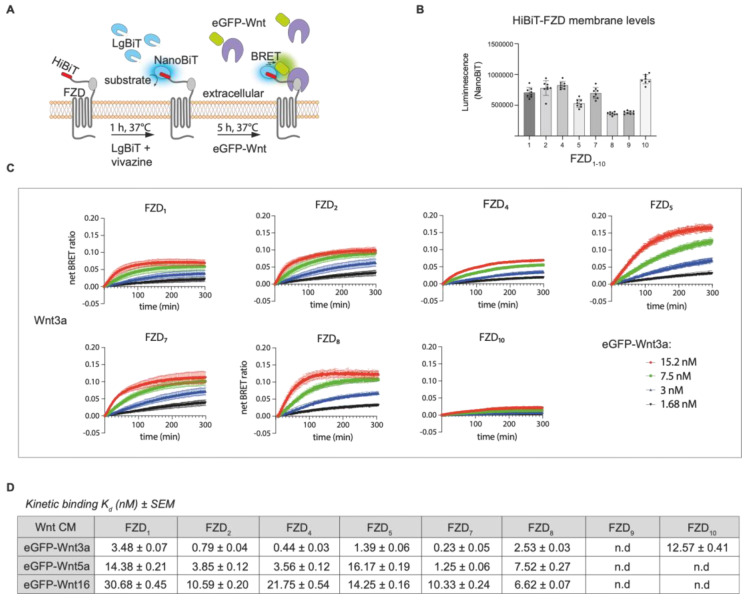
Comparison of Wnt3a binding to multiple FZD-expressing cell lines. (**A**) Schematic depiction of NanoBiT/BRET assay. (**B**) Estimation of relative HiBiT-FZD cell surface levels in FACS-sorted U-2 OS cells used for the eGFP-Wnt3a assay. (**C**) Association kinetics of eGFP-Wnt3a with U-2 OS cells expressing the indicated HiBiT-FZD proteins. Binding curves were generated over time using 1.5, 3, 7.5 and 15.2 nM of eGFP-Wnt3a. BRET ratios were measured every 80 s for a total of 5 h. Raw data were fitted to the ‘two or more hot concentrations model’ and are presented as mean ± SEM from n = 3–4 individual experiments, each performed with duplicate samples. (**D**) Summary of the calculated binding affinities of eGFP-Wnt3a, eGFP-Wnt5a and eGFP-Wnt16 to the various HiBiT-FZD’s. *K*_d_ values are based on data from n = 3–4 individual experiments and shown as best-fit value ± SEM. n.d = not detected. Kruskal–Wallis’s rank sum test statistics were used to determine significance between different FZD samples within experiments and are shown in Appendix A, along with the binding curves and relative FZD_1–10_ receptor densities for eGFP-Wnt5a and eGFP-Wnt16 experiments.

**Figure 4 cells-14-00810-f004:**
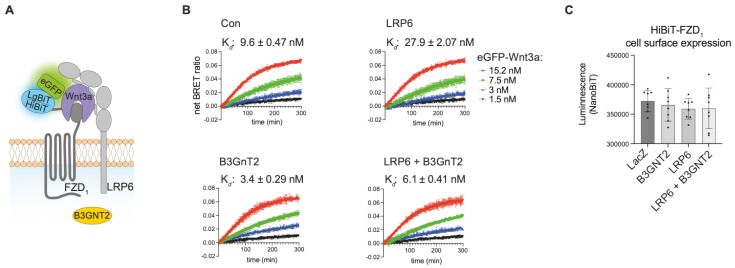
Influence of LRP6 on receptor ligand interaction. (**A**) Schematic drawing of trimeric HiBiT-FZD-eGFP-Wnt3a-LRP6 complex formation. (**B**) Association kinetics of eGFP-Wnt3a binding to U-2 OS cells stably expressing HiBiT-FZD_1_-mCherry. Cells were transfected with *LRP6*, *B3GNT2* or *LRP6* + *B3GNT2* as indicated, and 48 h post-transfection, the NanoBiT/BRET assay was performed. *LacZ* pDNA transfection was used for the control (Con) sample, and *LacZ* was also included in all samples for normalization purposes. Association kinetics were determined over time using 1.5, 3, 7.5 and 15.2 nM of eGFP-Wnt3a CM and measured every 80 s for 5 h. Raw data were fitted to the ‘two or more hot concentrations model’ and are presented as mean ± S.D from n = 3 individual experiments. (**C**) Cell surface expression of HiBiT-FZD_1_-mCherry after transfection with the indicated expression plasmids, as measured by NanoBiT luminescence.

## Data Availability

The original contributions presented in this study are included in the article/Appendix A. Further inquiries can be directed to the corresponding author.

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
