# Peer review of "Quantification of Wnt3a, Wnt5a and Wnt16 Binding to Multiple Frizzleds Under Physiological Conditions Using NanoBit/BRET"

_cells, 2025, doi:10.3390/cells14110810_

Round 1

Reviewer 1 Report

Comments and Suggestions for Authors

The manuscript "Quantification of Wnt3a, Wnt5a and Wnt16 Binding to Multiple Frizzleds Under Physiological Conditions using NanoBit/BRET" provides an update on the ongoing work in the laboratory to quantify Fzd-Wnt interactions in meaningful, quantitative ways. It is a welcome addition to the field.

Overall, the experimental quality is good, and the system demonstrated to a very high standard. 

One possible addition that would help the overall interest in the field would be an expansion on what the authors present in Figure 1. The images are quite small, but the asymmetric distribution of several Wnts is evident. This is mentioned in the text, but not further discussed in the discussion. The figure could be improved with the overlay images showing the asymmetry clearly.

Otherwise, the manuscript is fine. 

Reviewer 2 Report

Comments and Suggestions for Authors

Reviewer Comments for cells-cells-3610935
This manuscript presents a systematic analysis of Wnt ligand-Frizzled receptor interactions using a live-cell NanoBiT/BRET system by Dr. Wesslowski, et al. The authors used fluorescently tagged Wnt3a, Wnt5a, and Wnt16 to quantify real-time binding affinities to all ten Frizzled (FZD) receptors expressed at low physiological levels. Importantly, the study highlights the modulatory effect of LRP6 and its glycosylation by B3GnT2 on ligand-receptor affinity, providing insights into the trimeric Wnt-FZD-LRP6 complex. The methodological rigor and depth of data presented here represent a valuable resource for the Wnt signaling community. To further enhance the manuscript, the reviewer has some questions and suggestions that will contribute to its overall strength.
Major Comments and Suggestions:
•    Data Interpretation on FZD9:
Lines 341-344: The authors note a discrepancy between FZD9's strong signaling activity and its undetectable Wnt3a binding. This intriguing result warrants deeper discussion or experimental exploration—e.g., could post-binding events, receptor conformations, or co-receptor involvement explain this?
•    Statistical Treatment:
While error bars and sample sizes are described, the authors should include statistical comparisons (e.g., ANOVA or t-tests) for key conclusions, such as the differences in binding affinities between receptor types and with/without LRP6.
•    Figure Presentation:
Some figures, particularly the BRET kinetic curves (e.g., Figure 3C), are dense and could benefit from simplification. Showing selected curves in the main figure and moving complete datasets to the Supplementary Material may improve readability.
•    Supplementary Materials Clarity:
The supplementary figures support the main data well but need clearer figure legends. For example, the legends for Supplementary Figures 2 and 3 could better explain the rationale for the expression level categorizations and their downstream use.
Minor Comments:
•    Lines 21 and 318: Consider clarifying “low receptor levels” quantitatively.
•    Lines 374-376: FZD10 binding discrepancy vs previous study should be further contextualized—could expression levels or cellular context (U-2 OS vs HEK293) explain this?
•    Figure 4B: Consider adding binding curves for LacZ + B3GnT2 to better dissect B3GnT2’s independent effects.
•    Consistency: Use either "eGFP-Wnt3a" or "eGFP-WNT-3a" uniformly throughout text and figures (e.g., Lines 104 and 227).
•    Lines 293-296 say that co-expression of Wnt with Afamin increases the secretion of eGFP-Wnt6, 7a, 10a, and 10b. While the figure Supplementary 1A shows that there was a significant increase in the expression of other proteins as well, such as 1,2b and 16.
•    Lines 305 and 306 say that eGFP-Wnt16 also showed consistent association with different FZD, but the Supplementary Figure 1B shows that Wnt16 is the same as the others.
•    The experiment shown in Figure 1B is limited to Wnt3a, 51, and 16, confirming that they retained functional activity. What about other Wnt proteins? Did they not retain the functional activity, or were they not tested at all?
